# Multimodal Treatment Strategies to Improve the Prognosis of Locally Advanced Thoracic Esophageal Squamous Cell Carcinoma: A Narrative Review

**DOI:** 10.3390/cancers15010010

**Published:** 2022-12-20

**Authors:** Tadashi Higuchi, Yoshiaki Shoji, Kazuo Koyanagi, Kohei Tajima, Kohei Kanamori, Mika Ogimi, Kentaro Yatabe, Yamato Ninomiya, Miho Yamamoto, Akihito Kazuno, Kazuhito Nabeshima, Kenji Nakamura

**Affiliations:** Department of Gastroenterological Surgery, Tokai University School of Medicine, Isehara 259-1193, Japan

**Keywords:** esophageal squamous cell carcinoma, induction therapy, chemoradiation therapy, conversion surgery, salvage surgery

## Abstract

**Simple Summary:**

Esophageal squamous cell carcinoma with invasion to other organs/structures is considered unresectable, and its prognosis is extremely poor. In Japan, definitive chemoradiotherapy is often selected for cT4b cases, and regarded as the standard treatment option. In recent years, conversion surgery has been applied in some cases after downstage by induction therapy. In particular, DCF therapy consisting of docetaxel, cisplatin, and 5-fluorouracil has been reported effective for cT4b cases. Induction chemoradiotherapy, followed by conversion surgery, and salvage surgery after definitive chemoradiotherapy, have also been performed in selected patients. This article provides an overview of the contribution of multimodal treatment to improve oncological and survival outcomes for patients with locally advanced unresectable esophageal squamous cell carcinoma.

**Abstract:**

Esophageal cancer is the seventh most common malignancy and sixth most common cause of cancer-related death globally. Esophageal squamous cell carcinoma (ESCC) with aortic or tracheal invasion is considered unresectable, and has an extremely poor prognosis; its standard treatment is definitive chemoradiotherapy (dCRT). In recent years, induction chemotherapy (ICT) has been reported to yield high response rates for locally advanced ESCC, and the efficacy and safety of ICT followed by conversion surgery (CS) have been investigated. Multimodal treatment, combining surgery with induction chemoradiotherapy (ICRT) or ICT, is necessary to improve ESCC prognosis. CS is generally performed for locally advanced ECC after ICRT or ICT when tumor downstaging is achieved, although its prognostic benefit remains controversial. The Japan Clinical Oncology Group (JCOG) has conducted a three-arm phase III randomized controlled trial (JCOG1510) to confirm the superiority of DCF (docetaxel, cisplatin, and 5-fluorouracil) ICT, over conventional dCRT, among patients with initially unresectable ESCC. In recent years, researchers have reported favorable outcomes of induction therapy followed by CS and salvage surgery, after dCRT or systemic immunochemotherapy. In this review, we will describe the latest developments in the multimodal treatment including chemotherapy, CRT, surgery, and immunotherapy, which may improve oncological and survival outcomes for patients with cT4 ESCC.

## 1. Introduction

Esophageal cancer is considered as a highly malignant carcinoma in terms of its tumor biology. It tends to invade adjacent organs and structures, such as the trachea, bronchus, aorta, pericardium, and lung, due to the absence of a serous membrane [1]. According to the TNM classification [2], tracheal, bronchial, and aortic invasion are classified as T4b, and account for approximately 7% of all thoracic esophageal cancers [3]. Clinical T4b esophageal squamous cell carcinoma (ESCC) with invasion to other organs/structures is considered unresectable, and its prognosis is extremely poor. In Japan, definitive chemoradiation therapy (dCRT) is often selected for cT4b cases, and is regarded as the standard treatment [4,5,6,7,8,9]. In recent years, response rates to chemotherapy/CRT have improved, and conversion surgery (CS) has been performed when the primary tumor is downstaged by induction therapy. In particular, DCF therapy consisting of docetaxel (DTX), cisplatin (CDDP), and 5-fluorouracil (5-FU) has been reported to be effective for cT4b ESCC. The efficacy and safety of CS, after DCF as induction chemotherapy (ICT), has been studied [10,11,12,13,14,15]. A randomized controlled trial (JCOG1510) is now in progress to confirm the advantage of DCF ICT over conventional dCRT, regarding overall survival in patients with cT4b ESCC [16]. To improve cT4b ESCC prognosis, combination therapy of surgery with CRT or chemotherapy may be necessary. However, the safety and prognostic value of CS for patients who achieved tumor downstaging by induction therapy, and salvage surgery after dCRT or systemic immunochemotherapy, remain controversial.

This article provides an overview of the contribution of multidisciplinary treatment for cT4b ESCC to improve oncological and prognostic outcomes for those patients.

## 2. Materials and Methods

This article is a narrative review. Relevant articles were searched in PubMed using the following keywords “esophageal cancer”, “cT4b”, “induction chemotherapy”, “definitive chemoradiotherapy”, “salvage surgery”, and “conversion therapy”. Reference lists of the included studies and related comments were manually filtered to search for studies of possible relevance. Finally, randomized controlled trials and cohort studies were selected, excluding those that lacked necessary data including treatment and prognostic outcomes. Selected prospective clinical trials for cT4b esophageal cancer included in this review were mostly conducted in Japan.

## 3. Treatment Strategy for Locally Advanced Thoracic Esophageal Squamous Cell Carcinoma

T4b ESCC which invades directly to the aorta, trachea, bronchus, pulmonary veins, pulmonary artery, and/or vertebral body is considered unresectable. The treatment algorithm for cT4b ESCC in Japan is defined by the Esophageal cancer practice guidelines [17], following the Japanese Classification of Esophageal Cancer [18]. The standard treatment for cT4b ESCC without distant organ metastasis in Japan is dCRT comprising radiation (60 Gy) with CDDP (70 mg/m^2^ day 1) and 5-FU (700 mg/m^2^ day 1–4), according to the results of a phase II trial conducted in Japan [8]. However, CS has recently been reported for patients who have undergone ICT or induction CRT (ICRT).

### 3.1. Definitive Chemoradiation Therapy

From 1992 to 1994, a phase II trial of CRT with two courses of radiotherapy (30 Gy), followed by one course of CDDP+5-FU (CF) therapy was conducted by the Japan Clinical Oncology Group (JCOG) Esophageal Cancer Group [19]. Complete response (CR) rate was 11%, and the objective response rate (ORR) was 64% for 45 patients enrolled in the study. Subsequently, a phase II trial for concurrent chemoradiation with CF therapy in T4 patients was conducted (JCOG9516) [8], resulting in a CR rate of 15%. In another phase II trial by Ohtsu et al. [5], CRT with radiotherapy (60 Gy) and CF *was administered for* T4 and/or M1 (lymph node) patients, and the CR rate was 33%. These results suggest that CRT is an effective curative treatment option for Stage IV, including cT4b ESCC, in patients with good performance status. As summarized in Table 1 and Table 2, seven prospective studies examined the outcome of dCRT for patients with cT4b ESCC in Japan.

#### 3.1.1. Definitive Chemoradiation Regimen

Nishimura et al. [20] conducted a prospective study on the long-term administration of 5-FU (300 mg/m^2^) and CDDP (10 mg/body) over 2 weeks, combined with 60 Gy radiotherapy over 7 weeks for T4 esophageal cancer. The long-term treatment was intended to maximize radiosensitivity, as well as to avoid rapid tumor shrinkage. Of the 25 patients enrolled in this study, CR was achieved in 8 (25%) patients. Based on the results, compared the previous JCOG9516 trial, the current standard regimen for concurrent chemotherapy in both Japan and the U.S. is CDDP: 70–75 mg/m^2^, followed by 5-FU: 700–1000 mg/m^2^/day for four days. JCOG0303 [7] was a clinical trial that examined the superiority of a low-dose fractionated CF regimen over standard CF therapy; however, the median survival time was similar, and there was no significant difference in adverse events. Based on these results, the standard treatment for locally advanced unresectable ESCC in Japan is dCRT with CF therapy (CDDP: 70 mg/m^2^ day 1, 5-FU: 700 mg/m^2^ day 1–4, two courses every 4 weeks, radiation total 60 Gy/30 Fr). In general, CF therapy is the most frequently used chemotherapeutic regimen in combination with radiation therapy as dCRT; however, the DCF regimen has recently been reported to improve long-term results [21,22].

#### 3.1.2. Treatment Response and Prognosis for dCRT

CR rates and ORR of the clinical trials on dCRT for cT4 esophageal cancer included in this review (Table 1 and Table 2) were 0–60% and 68–88%, respectively. Prolonged prognosis was observed for CR cases compared to non-CR cases, with a 3-year overall survival (OS) in the range of 23–79%. Ohtsu et al. [5] performed a single-center study for radiation (60 Gy) combined with CF therapy in 54 patients with T4 and/or M1 (lymph node), which resulted in ORR 87%, 2-year OS rate 23%, and a median survival time (MST) of 9 months. Similar results were reported in another study by Ishida et al. [8], a multicenter phase II study of 60 patients who underwent dCRT for a similar cohort, with ORR 68%, 2-year OS rate 31.5%, and MST 10 months. Other studies for dCRT against T4 ESCC have also reported similar results, with CR rates in the range of 15–33% and 3-year OS of 23–26% [6,24]. In recent studies utilizing DCF combined with radiation therapy for T4 esophageal cancer, reported CR rates ranged from 48 to 60% [21,22], which seems highly effective compared to conventional dCRT, with acceptable adverse events. Continuous surveillance is necessary even for CR cases since a certain number of ESCC patients suffer long-term recurrence after dCRT.

#### 3.1.3. Adverse Events and Mortality after dCRT

An esophageal fistula is one of the most fatal treatment-related complications after CRT for T4b esophageal cancer; its incidence was reported to be as high as 22% in the JCOG0303 study [7]. The reported incidence of severe treatment-related complications after CRT for cT4b cases, including tracheoesophageal fistula and esophageal perforation, is 10–20% [5,6,7,25]. Clinical trials on CRT for advanced esophageal cancer often exclude cases with fistula formation; therefore, the utility of CRT for patients with a fistula remains controversial. Nishimura et al. [20] reported that when CRT was administered for five T4 esophageal cancers with fistulas, two patients discontinued treatment due to worsening of the fistula, and there was one treatment-related death. In contrast, in two of the remaining three cases, the fistula was cured, and those patients were able to take food orally afterward. Thus, CRT for advanced esophageal cancer with fistula formation may be performed in selected cases; however, exacerbation of the fistula should always be considered.

Hematologic toxicity is one of the most frequently reported adverse events after CRT, with decreased white blood cell and neutrophil counts being the most common. Ohtsu et al. [5] reported that severe anemia and decreased white blood cell/platelet counts occurred in 28%, 24%, and 17% of the patients who received CRT, respectively, with a treatment-related mortality of 7%. Ishida et al. [8], similarly, reported severe white blood cell decrease in 33% of the patients, with treatment-related deaths in 3% of the patients. Notably, the incidence of decreased white blood cell and neutrophil counts in CRT utilizing DCF was much higher than those after conventional CRT (71–92% and 57–76%, respectively) [21,22,23].

As indicated in Table 2, other common adverse events associated with dCRT include esophagitis (15–29%) and anorexia (12–21%).

### 3.2. Induction Chemotherapy

Although the standard of care for unresectable T4 ESCC is CRT, a number of clinical trials have evaluated the efficacy of ICT followed by CS since the 1990s. In particular, DCF therapy has been reported effective for cT4b cases, and DCF ICT combined with CS is now regarded as an alternative treatment option for T4 cases.

#### 3.2.1. Induction Chemotherapy Regimen

In the 1990s, Ancona et al. [26] reported a downstage rate of 62%, R0 resection rate of 32%, and 5-year OS rate of 11% for CF therapy as ICT on T4 esophageal cancer. Since then, ICT in T4 cases utilizing CF therapy have not been reported. One of the reasons for this phenomenon may be the insufficient effect of CF therapy as ICT. The response rate of CF therapy, which has been considered as the standard chemotherapeutic regimen, has been reported as 35%, which is lower than that of CRT in terms of local control [27,28]. In recent years, DCF therapy has attracted attention as an induction treatment for locally advanced esophageal cancer [23,29]. In a phase III JCOG1109 trial for resectable locally advanced esophageal cancer [30], pathological CR rate was higher for DCF compared to CF therapy (19.8% vs. 2.1%), with a manageable toxicity profile. The COSMOS trial [12], conducted in Japan, is a multicenter Phase II study, which evaluated the safety and efficacy of DCF therapy for T4 ESCC. The primary endpoint was 1-year OS, which was 68% in patients who achieved tumor downstaging and underwent CS. There were no mortalities or serious complications related to surgery, demonstrating the safety and efficacy of CS after DCF ICT for T4 ESCC. A Phase III JCOG1510 study [16], conducted to verify the superiority of DCF ICT over conventional dCRT for locally advanced unresectable thoracic esophageal cancer is now undergoing. In this study, patients who were diagnosed as operable after DCF ICT will undergo CS, whereas unresectable patients will receive additional treatment by dCRT. The result of this study is awaited to confirm the efficacy and safety of DCF ICT. The current regimen of DCF therapy as ICT is generally 2–3 courses of 5-FU: 700–750 mg/m^2^, CDDP: 70–75 mg/m^2^, and DTX: 70 mg/m^2^ [12,23,29].

#### 3.2.2. Toxicity and Mortality after ICT

The results of the COSMOS trial [12] showed that the main adverse events of DCF ICT were white blood cell/neutrophil decrease, which were observed in 42% and 66% of patients, respectively, indicating high hematologic toxicity. In addition, febrile neutropenia was observed in 23% of the patients. Non-hematological adverse events included anorexia (25%), diarrhea (10%), and esophageal fistula (4.2%); however, there were no treatment-related deaths. Although related to high hematologic and non-hematologic toxicity, appropriate management may prevent serious complications during DCF ICT.

### 3.3. Induction Chemoradiotherapy

Along with ICT, ICRT may also lead to tumor downstaging and enable R0 resection. Reported ORR/CS rates are as high as 60–76%/35–84%, respectively [31,32,33,34,35]. Ikeda et al. [31] reported that CF therapy combined with radiation (30Gy) lead to a cCR rate of 24%. CS for patients with ICRT-induced downstaging should be regarded as a different treatment strategy from salvage surgery after dCRT, as discussed below.

#### 3.3.1. Induction Chemoradiotherapy Regimen

In 2005, Fujita et al. [36] conducted a prospective cohort study comparing ICRT and postoperative chemoradiotherapy for patients diagnosed with T4b esophageal cancer. ICRT consisted of radiation (36Gy) and CF therapy, with an additional postoperative 24 Gy radiation. Downstaging was achieved in 73% of the patients, and R0 resection was possible in 42% of the patients. Previous reports for T4 ESCC suggest that CRT at 40–60 Gy improves survival outcomes [37,38]. Additionally, 30–40 Gy is the most common dose for radiation in ICRT, which is lower than that of dCRT, to reduce complications. The chemotherapy used in ICRT is mainly CF therapy (5-FU: 500–1000 mg/m^2^, CDDP: 24–75 mg/m^2^), but the dosage varies depending on the trial.

#### 3.3.2. Response to ICRT and Patient Prognosis

Some reports indicated that the therapeutic effect of CRT for esophageal cancer is closely related to the invasion depth of the primary tumor, and CR is often not obtained in T4 esophageal cancer [39,40]. Fujita et al. [36] reported an ORR of 73%, CS rate of 58%, R0 resection rate of 42%, and pCR of 4% for the patients who underwent ICRT. Ikeda et al. [31] similarly reported an ORR of 76%, CS rate of 35%, R0 resection rate of 32%, and pCR of 3%. Analysis in the former study has indicated that the prognosis of T4 esophageal cancer patients was influenced by the therapeutic effect of ICRT, but the presence of CS was not directly related to survival outcomes; therefore, the survival benefit of additional CS remains unclear. Ishikawa et al. [38] studied 40 cases of T4b initially unresectable esophageal cancer, and reported R0 resection rate of 60% and pCR of 28% with 50 Gy radiation and CF therapy as ICRT. Yano et al. [39] also reported that the R0 resection rate was 42% and pCR was 29% with 40–60 Gy radiation and CF therapy, suggesting that higher radiation dose may lead to higher R0 resection and pCR rates, and thus may lead to better patient prognosis. In the recent phase III trial for resectable locally advanced esophageal cancer [30], a higher pCR rate was achieved by preoperative CRT (radiation 41.4Gy plus CF therapy, 38.5%) compared to preoperative chemotherapy (DCF therapy, 19.8%,), however, comparable R0 resection rate (87.5% vs. 85.6%) was observed. The treatment and prognostic impact of ICRT versus DCF ICT followed by CS for T4 esophageal cancer remain unknown, thereby the long-term results of the JCOG1510 study and a phase II study comparing DCF ICT vs. ICRT [29], discussed below, are awaited.

#### 3.3.3. Toxicity and Mortality after ICRT

The major adverse events were hematological toxicity, with severe anemia, white blood cell, and platelet decrease in 35%, 12%, and 8% of the patients, respectively, in the report by Fujita et al. [36]. They also reported the incidence of esophageal fistula as 8%. In the report by Ikeda et al. [31], severe adverse events included anemia and neutrophil decrease in 14%, diarrhea in 5%, and esophageal fistula in 4% of patients, and treatment-related death in one patient. In the COSMOS trial [12], a high incidence rate of hematologic toxicity was also reported, with anemia, white blood cell, and neutrophil decrease occurring in 11%, 28%, and 6% of the patients, respectively. Esophageal fistula was of concern as a treatment-related complication for ICRT in T4b ESCC; however, its incidence was comparable to ICT, and lower than that of dCRT [29]. This may be due to the reduction in the irradiation dose in ICRT compared to dCRT. In the JCOG 1109 study [30] for resectable thoracic esophageal cancer, preoperative CRT was related to a higher rate of esophagitis (8.9% vs. 1.0%) but lower febrile neutropenia (4.7% vs. 16.3%), compared to preoperative DCF. Incidence of treatment-related death was comparable with conventional preoperative CF therapy for both CRT and DCF, demonstrating the safety of either treatment under correct management.

## 4. Surgical Treatment of cT4b ESCC

The surgical strategy for locally advanced unresectable ESCC includes CS after ICT or ICRT, and salvage surgery after dCRT. Salvage surgery is often related to a high postoperative complication rate, as discussed below, and should be carefully considered when applying it to cT4b ESCC patients.

### 4.1. Salvage Surgery following dCRT

Swisher et al. [40] were the first to report on salvage surgery for esophageal cancer patients. Here, salvage surgery was defined as “esophagectomy for patients with local recurrence after dCRT”. In Japan, salvage surgery was defined as “surgery for residual cancer or recurrence after definitive (chemo) radiation therapy, with a radiation dose of 50 Gy or more” in the Japanese Classification of Esophageal Cancer, tenth edition [41]. In a retrospective study from Italy [42], 96.1% of the 51 eligible patients underwent surgical treatment after CRT, of which 39.2% of the patients achieved R0 surgery. Patients, after R0 resection, were reported to have better prognosis compared to non-curative resection. The MST was 11.1 months, 3-year OS rate was 8.8%, 5-year OS rate was 5.9%, while surgery-related mortality was as high as 10.2%. In a Japanese report [31], 37 T4 esophageal cancer patients were followed up after CRT, and only 13 patients with response to chemoradiotherapy underwent surgery, 12 of whom underwent R0 surgery. Better survival outcomes were observed for patients who underwent salvage surgery, compared to unresected cases. Tachimori et al. [43] reported that the 3-year OS rates for the 54 patients who underwent salvage surgery were 37.8%, with a prolonged prognosis seen in patients who underwent R0 resection. Frequently reported complications after salvage surgery after dCRT include anastomotic leakage in 14–38%, and respiratory complications in 6–62%. Previous reports have indicated that postoperative complications are more likely to occur after salvage surgery, compared to surgery alone or surgery after neoadjuvant therapy [44,45,46,47,48,49,50,51,52,53,54,55,56]. In a single-arm confirmatory study for salvage surgery after dCRT for Stage II/III ESCC, excluding T4 cases [57], 27% of the 94 patients included in the efficacy analysis underwent salvage surgery, and curative resection was achieved in 76% of those patients. Early and late severe postoperative complications were observed in 20 and 9.6% of the salvage cases, respectively. Survival analysis showed a favorable outcome, in which 3-year OS was 74.2% for the entire cohort. Salvage surgery may be applied for selected patients, and the development of preoperative measures to predict R0 resection is needed.

### 4.2. Conversion Surgery following Induction Therapy

As discussed above, recent advances of ICT and ICRT enabled resection of initially unresectable ESCC for patients who responded to induction therapy. The term “conversion surgery” is generally used for surgery against tumors which were initially unresectable, but became resectable by tumor downstaging. Table 3 and Table 4 show the results of prospective studies of CS for cT4b ESCC in Japan. In the phase II COSMOS study [12], 42% of the 48 patients underwent esophagectomy after DCF ICT, with an R0 resection rate of 40%, and a favorable 1-year OS rate which was 67.9%. Prolonged prognosis was expected for patients who underwent induction therapy, specifically those who achieved downstaging and were eligible for R0 resection. In total, 47.9% of the patients became cancer-free after R0 resection by CS, or CR by DCF ICT. Postoperative complications included recurrent laryngeal nerve palsy (RLNP), and respiratory infections in 38.1 and 14.3% of the patients, respectively. Regarding ICRT plus CS, Ikeda et al. [31] reported a CS rate of 35%, and R0 resection rate of 32% for T4 esophageal cancer. Favorable 1- and 5-year OS were observed for patients who underwent CS (83 and 57%, respectively). In the report by Fujita et al. [36], CS was performed for 15 out of 26 patients (58%), with a complete resection rate of 42%. Frequently observed complications after CS were RLNP, pneumonia, and tracheal ischemia, which occurred in 60%, 22%, and 27% of the patients, respectively. Importantly, survival outcomes were significantly better for the patients who achieved response to ICRT than those who did not; however, there were no difference in 5-year OS between CS and non-resected cases. In a recent multicenter randomized phase II trial, ICRT and DCF ICT were compared to verify the optimal induction therapy for cT4b esophageal cancer [29]. Short-term results indicated that the CS, R0 resection, and postoperative complication rates were similar for both groups; however, adverse events of the initial induction therapy were more frequently observed in the DCF ICT group, and the histological CR rate was higher in the ICRT group. The combination of chemotherapy, radiation therapy, and surgical modalities may enable curative resection for initially unresectable locally advanced ESCC. The value of CS should be further discussed, and long-term results of the randomized clinical trials are awaited to decide on the optimal treatment strategy for T4 cases.

## 5. Immune Checkpoint Inhibitor for Unresectable Advanced Esophageal Cancer

As a result of recent global multicenter, randomized, phase III trials [58,59], an immune checkpoint PD-1 inhibitor in combination with chemotherapy became one of the first-line treatment options for unresectable advanced ESCC. In the KEYNOTE-590 trial [58], 29 patients (8%) with unresectable locally advanced disease underwent pembrolizumab plus chemotherapy, and the ORR for the entire cohort who underwent pembrolizumab plus chemotherapy was 45%. In the CheckMate 648 trial [59], 44 patients (14%) of the unresectable advanced patients underwent nivolumab plus chemotherapy, with an ORR of 42% for the entire cohort. In the subgroup analysis, nivolumab plus chemotherapy demonstrated relatively higher (but not significant) OS when compared to chemotherapy alone, in the patients with unresectable advanced esophageal cancer (MST, 12.8 vs. 12.1 m; hazard ratio, 0.92; 95% confidence interval, 0.45–1.16). Among the 321 patients in the nivolumab plus chemotherapy group, 9 patients underwent additional surgical treatment, whereas only 1 patient achieved curative resection. In a report by China [60], a preoperative PD-1 inhibitor plus chemotherapy were compared to chemotherapy alone for patients with advanced ESCC, including 102 (65.8%) cT4b cases. ORR and pathological CR rate were significantly higher for the PD-1 inhibitor plus chemotherapy group, with an acceptable postoperative complication rate. The safety and efficacy of induction immunochemotherapy against DCF ICT or ICRT remain unclear; thus, further studies are awaited.

## 6. Conclusions

With recent advances in the induction therapy regimen and patient management, conversion surgery for patients with initially unresectable thoracic esophageal cancer has become more common. Although conversion surgery is often related to a higher complication rate compared to surgery alone or surgery after neoadjuvant treatment, complications are usually manageable. A recent study has shown that CS may be an effective treatment option when curative resection is achieved; therefore, the development of a more effective induction therapy is needed, and thus the results of the current ongoing and future clinical trials are awaited. Pending the outcome of the ongoing clinical trials, surgeons, radiologists, and medical oncologists are encouraged to discuss the choice of an appropriate treatment strategy for each individual case, from a safety and efficacy standpoint.

## Figures and Tables

**Table 1 cancers-15-00010-t001:** Clinical trials on definitive chemoradiotherapy for clinical T4 esophageal cancer.

No.	Author	Year	Design	Histology	N	Radiation (Gy)	Chemotherapy Regimen		
							DTX	CDDP	5-FU
1	Ohtsu et al. [5]	1999	Phase II	SCC	54	60		40 mg/m^2^	400 mg/m^2^
2	Nishimura et al. [20]	2002	Phase I/II	SCC	28	60		10 mg/body	300 mg/m^2^
3	Ishida et al. [8]	2004	Phase II (JCOG9516)	SCC	60	60		70 mg/m^2^	700 mg/m^2^
4	Higuchi et al. [21]	2014	Phase II	SCC	42	50.4/61.2	35 mg/m^2^	40 mg/m^2^	400 mg/m^2^
5	Shinoda et al. [7]	2015	Phase II (JCOG0303)	SCC	68	60		4 mg/m^2^	200 mg/m^2^
					71	60		70 mg/m^2^	700 mg/m^2^
6	Miyazaki et al. [22]	2015	Phase I/II	SCC/AC	37	60	50 mg/m^2^	60 mg/m^2^	600 mg/m^2^
7	Satake et al. [23]	2016	Phase I/II	SCC	33	60	70 mg/m^2^	70 mg/m^2^	750 mg/m^2^

JCOG, Japan Clinical Oncology Group; SCC, squamous cell carcinoma; AC, adenocarcinoma; DTX, docetaxel; CDDP, cisplatin; 5FU, 5-fluorouracil.

**Table 2 cancers-15-00010-t002:** Results of the clinical trials indicated in Table 1.

No.	Author	Adverse Events (CTCAE v5.0 Grade 3/4)		Fistula Formation	Mortality	ORR	cCR Rate	MST	OS
1	Ohtsu et al. [5]	Anemia	28%	14%	7%	81%	25%	9 m	3-year: 23%
		WBC decreased	24%						
		PLT decreased	17%						
		Esophagitis	15%						
2	Nishimura et al. [20]	Anemia	21%	18%	7%	88%	32%	5 m	2-year: 23%
		WBC decreased	50%						
		PLT decreased	11%						
		Nausea	32%						
3	Ishida et al. [8]	WBC decreased	33%	N/A	3%	68%	15%	10 m	2-year: 32%
		AST/ALT increased	15%						
4	Higuchi et al. [21]	WBC decreased	71%	5%	2%	86%	30/60% ^†^	29 m	3-year: 44%
		NEU decreased	57%						
		Febrile neutropenia	38%						
		Esophagitis	29%						
5	Shinoda et al. [7]	Anemia	6/14% *	22/18% *	1/3% *	N/A	1/0% *	14/13 m *	3-year: 26/26% *
		WBC decreased	21/26% *						
		NEU decreased	7/19% *						
		Hyponatremia	13/17% *						
		Anorexia	21/20% *						
6	Miyazaki et al. [22]	WBC decreased	92%	5%	0%	86%	48%	53 m	3-year: 44%
		NEU decreased	76%						
		Febrile neutropenia	22%						
		Nausea	32%						
		Esophagitis	24%						
7	Satake et al. [23]	WBC decreased	24%	6%	0%	73%	39%	26	3-year: 79%
		NEU decreased	18%						
		Anorexia	12%						

CTCAE v5.0, Common Terminology Criteria for Adverse Events version 5.0; ORR, overall response rate; CR, complete response; MST, median survival time; OS, overall survival; WBC, white blood cell; PLT, platelet; NEU, neutrophil; AST, aspartate aminotransferase; ALT, alanine aminotransferase. * Indicated for low-dose/high-dose CF therapy. ^†^ Indicated for 50.4/61.2 Gy.

**Table 3 cancers-15-00010-t003:** Clinical trials on conversion surgery for clinical T4 esophageal cancer.

No.	Author	Year	Design	Histology	N	Radiation (Gy)	Chemotherapy Regimen
1	Ikeda et al. [31]	2001	Phase II	SCC	37	30	CDDP: 70 mg/m^2^, 5-FU: 700 mg/m^2^
2	Fujita et al. [36]	2005	Cohort	SCC	26	36	CDDP: 24 mg/m^2^, 5-FU: 500 mg/body
3	Shimoji et al. [51]	2013	Cohort	SCC/AC	87	40–66	CDGP: 90 mg/m^2^, 5-FU: 900 mg/m^2^
						-	CDGP: 50 mg/m^2^, ADM: 30 mg/m^2^, 5-FU: 600 mg/m^2^
4	Yokota et al. [12]	2016	Phase II	SCC	48	-	DTX: 70 mg/m^2^, CDDP: 70 mg/m^2^, 5-FU: 750 mg/m^2^
5	Sugimura [29]	2021	Phase II	SCC	99	50.4	CDDP: 75 mg/m^2^, 5-FU: 1000 mg/m^2^
							DTX: 70 mg/m^2^, CDDP: 70 mg/m^2^, 5-FU: 700 mg/m^2^

SCC, squamous cell carcinoma; AC, adenocarcinoma; 5FU, 5-fluorouracil; CDDP, cisplatin; CDGP, nedaplatin; ADM, adriamycin; DTX, docetaxel.

**Table 4 cancers-15-00010-t004:** Results of clinical trials indicated in Table 3.

No.	Author	CS Rate	R0 Rate	pCR Rate	Postoperative Complications		Mortality	MST	OS
1	Ikeda et al. [34]	35%	32%	3%	Pneumonia	38%	3%	10 m	5-year: 23%
					Anastomotic leakage	15%			
2	Fujita et al. [39]	58%	42%	4%	Pneumonia	33%	7%	22 m	3-year: 23%
					RLNP	60%			
					Tracheal ischemia	27%			
					SBO	20%			
3	Shimoji et al. [53]	70%	61%	14%	Pneumonia	33%	14%	10/26 m *	5-year: 35%
					Anastomotic leakage	9%			
					Surgical site infection	21%			
4	Yokota et al. [12]	42%	40%	20%	Pneumonia	14%	0%	N/A	1-year: 68%
					RLNP	38%			
					Pleural effusion	24%			
					Esophagitis	29%			
5	Sugimura et al. [32]	84%/84% ^†^	78%/76% ^†^	40%/17% ^†^	Pneumonia	18%/37% ^†^	0/0% ^†^	N/A	N/A
					Anastomotic leakage	13%/10% ^†^			
					RLNP	18%/12% ^†^			

CS, conversion surgery; CR, complete response; RLNP, recurrent laryngeal nerve paralysis; SBO, small bowel obstruction; MST, median survival time; OS, overall survival; N/A, not available. * Indicated for CDGP+5-FU/CDGP+ADM+5-FU therapy. ^†^ Indicated for ICRT/ICT.

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
