# Peer review of "Multimodal Treatment Strategies to Improve the Prognosis of Locally Advanced Thoracic Esophageal Squamous Cell Carcinoma: A Narrative Review"

_cancers, 2022, doi:10.3390/cancers15010010_

Round 1
Reviewer 1 Report (Previous Reviewer 2)
i have no further suggestions
Author Response
Thank you so much for your kind review. Please see the attachment for point-by point response to the editor and reviewers' comments.

Reviewer 2 Report (New Reviewer)
Dear Editor and Esteemed Authors,
It was my pleasure to review your work titled “Multimodal treatment strategies to improve the prognosis of locally advanced thoracic esophageal squamous cell carcinoma: A narrative review” by Dr. Higuchi and colleagues from the Department of Gastroenterological Surgery, at Tokai University School of Medicine in Isehara, Japan.
In this quite thorough review paper the authors present an overview of treatment and management of patients with advanced thoracic esophageal squamous cell carcinoma and namely those with cT4b disease which means it invaded adjacent structures. The different benefit and outcomes of classical management, dCRT, induction chemo or induction RT followed by surgical resection in patients which responded and were downstaged is presented.
This is a very well constructed review and I am well impressed. It is thorough, methodical and presents and explains all different management strategies and available modalities of treatment. It presents a quite large body of evidence in an organized and easy to understand way. The language of the manuscript is clear and understandable with very minor and few English errors.
Overall I am quite keen to recommend outright acceptance apart from a minor point which is as follows:
1. In the simple summary I would change the first sentence and especially the term “invasion of other organs” to invasion of adjacent structures! Similarly in line 47 (an wherever else it is been used) I would change the term organs to structures (or add both terms i.e. organs/structures) because I would not call the trachea or the aorta an organ by itself!
If the authors address this (which is quite simple to do really) I would be happy to recommend acceptance.
Author Response
Thank you so much for your kind review. Please see the attachment for point-by point response to the editor and reviewers' comments.

This manuscript is a resubmission of an earlier submission. The following is a list of the peer review reports and author responses from that submission.
Round 1
Reviewer 1 Report
Some minor concernings to improve the quality:
- line 41, Trans Oral Robotic Surgery (TORS) is a modality in the management of head and neck surgery. Selective Neck Dissection (SND) is oncological safe procedure even in patients with lymph node metastases. The indication should be accurately discussed by the multidisciplinary tumor board, especially for salvage surgery. please cite doi:10.1016/j.anl.2021.05.007
- Patients with advanced esophageal cancer have a poor prognosis and limited treatment options after first-line chemotherapy, To improve cT4b ESCC’s prognosis and combination therapy of surgery with CRT or ICT may be necessary. Pembrolizumab prolonged OS versus chemotherapy as second-line therapy for advanced esophageal cancer in patients with PD-L1 CPS ≥ 10, with fewer treatment-related adverse events. cite doi:10.1200/JCO.20.01888
- A multicenter randomized prospective phase 2 trial of chemoradiotherapy (CRT) versus chemotherapy (CT) as initial induction therapy for conversion surgery (CS) in clinical T4b esophageal cancer was conducted, comparing treatment effects and adverse events (AEs). In Group A, CS was performed in 34 (69%) and 7 patients (14%) after initial and secondary treatment. In Group B, CS was performed in 25 (50%) and 17 patients (34%) after initial and secondary treatment. The R0 resection rate after initial and secondary treatment was similar (78% vs 76%, P = 1.000). AEs including leukopenia, neutropenia, febrile neutropenia, and diarrhea were significantly more frequent in Group B. Group A had better histological complete response of the primary tumor (40% vs 17%, P = 0.028) and histological nodal status (P = 0.038). cite doi:10.1097/SLA.0000000000004564
- The structure of the paper should be improved. Is this a systematic review? a narrative review? add a methods section. Specifiy retrievement process. Add a flow diagram PRISMA.
- Change also the title specificing the review type
Author Response
May 8, 2022
Professor Bruce Sun
Assistant Editor
Cancers
Dear Prof. Bruce Sun,
I would like to submit the revised manuscript entitled “Multimodal treatment strategies to improve the prognosis of locally advanced thoracic esophageal squamous cell carcinoma: A narrative review” (Manuscript No. cancers-1684207) for consideration for publication in Cancers as a Review Article. And I would like to submit our response to the comments point-by-point in the another file by using underline of text to show the changes.
I sincerely hope that you will find this manuscript acceptable for publication in the Cancers and that you will give favorable consideration to the article.
Best regards,
Kazuo Koyanagi, MD, PhD, FACS
Professor and Chairman
Department of Gastroenterological Surgery,
Tokai University School of Medicine
Tel: +81-463-93-1122
Fax: +81-463-95-6491
Email: kkoyanagi@tsc.u-tokai.ac.jp
List of the revisions and responses to the Reviewer’s suggestions
Response to Reviewer 1
Thank you very much for reviewing our manuscript and offering valuable comments.
We have provided point-by-point responses to your comments, and revised the
relevant portions of the manuscript.
- The structure of the paper should be improved. Is this a systematic review? a narrative review? add a methods section. Specifiy retrievement process. Add a flow diagram PRISMA.
We have added the following sentences in our revised manuscript (Page 2, Lines 65-74): “Materials and Method: This article is narrative review and not a systematic review. We searched for relevant articles in PubMed using the following keywords “esophageal cancer”, “cT4b”, “induction chemotherapy”, “definitive chemoradiotherapy”, “salvage surgery”, and “conversion therapy”. Also, the reference lists of the included studies and related comments were manually filtered to search for new studies of possible relevance. Articles were selected from thos in which randomized controlled trials (RCT) and cohort studies were conducted, excluding those that lacked necessary data including treatment outcomes, prognosis data, among others. The selected prospective trials of cT4b for esophageal cancer were all conducted in Japan.”
”
- Change also the title specificing the review type
As you pointed out, we have changed the title in our revised manuscript: “Multimodal treatment strategies to improve the prognosis of locally advanced thoracic esophageal squamous cell carcinoma: A narrative review”.
We shall look forward to hearing from you regarding our submission and will be happy to respond to any further questions or comments that you might have.
Reviewer 2 Report
Manuscript reviewing surgical / treatment strategies and clinical trials for advanced ESCC. The manuscript is well written, but is primarily discusses Japanese guidelines and a specific subset of trials. For a review, this seems a bit odd. What were the inclusion and exclusion criteria to be considered for inclusion of a trail or study in this review? Is this a complete overview of trails and multimodal treatment strategies?
Author Response
May 8, 2022
Professor Bruce Sun
Assistant Editor
Cancers
Dear Prof. Bruce Sun,
I would like to submit the revised manuscript entitled “Multimodal treatment strategies to improve the prognosis of locally advanced thoracic esophageal squamous cell carcinoma: A narrative review” (Manuscript No. cancers-1684207) for consideration for publication in Cancers as a Review Article. And I would like to submit our response to the comments point-by-point in the another file by using underline of text to show the changes.
I sincerely hope that you will find this manuscript acceptable for publication in the Cancers and that you will give favorable consideration to the article.
Best regards,
Kazuo Koyanagi, MD, PhD, FACS
Professor and Chairman
Department of Gastroenterological Surgery,
Tokai University School of Medicine
Tel: +81-463-93-1122
Fax: +81-463-95-6491
Email: kkoyanagi@tsc.u-tokai.ac.jp
List of the revisions and responses to the Reviewer’s suggestions
Response to Reviewer 2
Thank you very much for reviewing our manuscript and offering valuable comments.
We have provided point-by-point responses to your comments, and revised the
relevant portions of the manuscript.
- Manuscript reviewing surgical / treatment strategies and clinical trials for advanced ESCC. The manuscript is well written, but is primarily discusses Japanese guidelines and a specific subset of trials. For a review, this seems a bit odd. What were the inclusion and exclusion criteria to be considered for inclusion of a trail or study in this review?
We have added the following sentences in our revised manuscript (Page 2, Lines 65-74): “Materials and Method: This article is narrative review and not a systematic review. We searched for relevant articles in PubMed using the following keywords “esophageal cancer”, “cT4b”, “induction chemotherapy”, “definitive chemoradiotherapy”, “salvage surgery”, and “conversion therapy”. Also, the reference lists of the included studies and related comments were manually filtered to search for new studies of possible relevance. Articles were selected from those in which randomized controlled trials (RCT) and cohort studies were conducted, excluding those that lacked necessary data including treatment outcomes, prognosis data, among others. The selected prospective trials of cT4b for esophageal cancer were all conducted in Japan.”
- Is this a complete overview of trails and multimodal treatment strategies?
As you pointed out, the content of this discussion is based on clinical trials in Japan. After reviewing the selected reference, the results are in according with the Japanese guidelines. Although we think that it covers the multimodal treatment strategies for cT4b esophageal cancer, we have changed the title in our revised manuscript: “Multimodal treatment strategies to improve the prognosis of locally advanced thoracic esophageal squamous cell carcinoma: A narrative review”.
We shall look forward to hearing from you regarding our submission and will be happy to respond to any further questions or comments that you might have.
Reviewer 3 Report
This review might focus on the multimodal treatments for LA-ESCC. Authors focus on induction chemotherapy and definitive CRT, but If authors discuss this title, authors need more information on the development of ICI-containing definitive CRT.
Major comments
- Authors might clarify treatments for "unresectable" locally advanced ESCC in the title.
- If authors focus on this theme, authors need to add information on the development of ICI-containing definitive CRT.
Minor comments
- L17:Would “ICT followed by conversion surgery” be the correct expression instead of “ICT after conversion surgery
Author Response
May 8, 2022
Professor Bruce Sun
Assistant Editor
Cancers
Dear Prof. Bruce Sun,
I would like to submit the revised manuscript entitled “Multimodal treatment strategies to improve the prognosis of locally advanced thoracic esophageal squamous cell carcinoma: A narrative review” (Manuscript No. cancers-1684207) for consideration for publication in Cancers as a Review Article. And I would like to submit our response to the comments point-by-point in the another file by using underline of text to show the changes.
I sincerely hope that you will find this manuscript acceptable for publication in the Cancers and that you will give favorable consideration to the article.
Best regards,
Kazuo Koyanagi, MD, PhD, FACS
Professor and Chairman
Department of Gastroenterological Surgery,
Tokai University School of Medicine
Tel: +81-463-93-1122
Fax: +81-463-95-6491
Email: kkoyanagi@tsc.u-tokai.ac.jp
List of the revisions and responses to the Reviewer’s suggestions
Response to Reviewer 3
Thank you very much for reviewing our manuscript and offering valuable comments.
We have provided point-by-point responses to your comments, and revised the
relevant portions of the manuscript.
- Authors might clarify treatments for "unresectable" locally advanced ESCC in the title. If authors focus on this theme, authors need to add information on the development of ICI-containing definitive CRT.
We have added the following sentences in our revised manuscript (Page 10, Lines 344-350): “4. Positioning of immune checkpoint inhibitor for unresectable advanced esophageal cancer: Recently, positive results of phase III trials with immune checkpoint inhibitors (ICI) have been reported for esophageal cancer [59]. In Japan, combination therapy with ICI and chemotherapy has become one of the first-line treatment options for unresectable advanced esophageal cancer. It is also suggested that chemotherapy and radiotherapy combined with ICI might be developed as treatment modality for cT4b ESCC.”
- L17:Would “ICT followed by conversion surgery” be the correct expression
instead of “ICT after conversion surgery
We have added the following sentences in our revised manuscript (Page 1, Lines 30):
“ICT followed by conversion surgery”
We shall look forward to hearing from you regarding our submission and will be happy to respond to any further questions or comments that you might have.
Round 2
Reviewer 2 Report
thank you for the clarification, i have no further suggestions
Reviewer 3 Report
Authors focus on conversion strategy for cT4b ESCC, in the world, not conversion strategy. but dCRT+ICI strategy is important and developing in some clinical trials.